# Effect of Magnetic Heating on Stability of Magnetic Colloids

**DOI:** 10.3390/nano12173064

**Published:** 2022-09-03

**Authors:** Andrzej Drzewiński, Maciej Marć, Wiktor W. Wolak, Mirosław R. Dudek

**Affiliations:** Institute of Physics, University of Zielona Góra, ul. Szafrana 4a, 65-069 Zielona Góra, Poland

**Keywords:** iron oxide nanoparticles, aqueous alkaline solutions, ultrasonic technique, magnetic heating

## Abstract

Stable aqueous suspension of magnetic nanoparticles is essential for effective magnetic hyperthermia and other applications of magnetic heating in an alternating magnetic field. However, the alternating magnetic field causes strong agglomeration of magnetic nanoparticles, and this can lead to undesirable phenomena that deteriorate the bulk magnetic properties of the material. It has been shown how this magnetic field influences the distribution of magnetic agglomerates in the suspension. When investigating the influence of the sonication treatment on magnetic colloids, it turned out that the hydrodynamic diameter as a function of sonication time appeared to have a power-law character. The effect of magnetic colloid ageing on magnetic heating was discussed as well. It was shown how properly applied ultrasonic treatment could significantly improve the stability of the colloid of magnetic nanoparticles, ultimately leading to an increase in heating efficiency. The optimal sonication time for the preparation of the magnetic suspension turned out to be time-limited, and increasing it did not improve the stability of the colloid. The obtained results are important for the development of new materials where magnetic colloids are used and in biomedical applications.

## 1. Introduction

The most common non-toxic superparamagnetic nanoparticles for various applications are iron-based magnetite (Fe_3_O_4_) nanoparticles. This material offers a wide range of applications due to its high energy absorption efficiency from the alternating magnetic field [1] and, as a result, it can be used for localized remote heating in the AC magnetic field treatment. The best known application of magnetic heating with nanoparticles is magnetic hyperthermia. However, the potential applications of magnetic heating are not limited to biomedical applications. They are widely studied in pollution removal, heat transfer, and tribology or to drive catalytic chemical reactions in bulk [2] or for a reaction near the surface of functional materials [3].

Among the various methods to synthesize iron oxide nanoparticles [4,5,6,7,8,9,10,11], the co-precipitation method is often chosen in the search for reliable and economical methods of synthesizing large quantities of nanoparticles. The applied method of co-precipitating nanoferrites from a precursor mixture of iron salts in ammonium hydroxide was derived from the technique proposed by Massart [12]. The disadvantage of prepared nanoparticles is their strong agglomeration and the related need to stabilize their suspensions in solutions [13], which makes their application much more difficult [14]. Since there is always a short-range attraction force between molecules, namely the Van der Waals force, which does not depend remarkably on the ionic strength, we always deal with its binding character. Moreover, the properties of ferrofluids significantly depend on the magnetic dipole-dipole attractions between magnetic nanoparticles, provided that they are close enough to each other [15]. For comparison, for a homogeneous suspension of iron oxide nanoparticles, where the particles are separated, the dipolar interactions between them are negligible [16].

When the nanoparticles are smaller than 50 nm, the stability of their aqueous suspension can be achieved by tuning the pH value of the solution. In the case of iron oxide magnetic nanoparticles (MNP), their size often does not exceed the value of 30 nm. In the aqueous environment, they acquire a pH-dependent surface charge: negative for pH above the point of zero charge (PZC), otherwise positive. This causes a repulsion to appear between them, counteracting the forces of magnetic attraction. The repulsive forces between molecules are due to the presence of their electrical double layers. To determine the stability of the suspension, the zeta potential is often used (the effective charge at the shear plane), which depends both on the magnitude of surface charge and the range of electrostatic interactions. Its value directly translates into the rate of agglomeration and sedimentation of particles. It is worth mentioning that the starting material for our research was bare magnetite nanoparticles with an average size of about 12 nm, forming a stable suspension in a solution with a pH = 11.5. Of course, taking into account medical applications, such a pH value is far from normal physiological conditions.

In our previous publication [17], we reported that the radio frequency magnetic field (RFMF) destabilizes a colloidal suspension of magnetic nanoparticle clusters. The reason was the aggregation of nanoparticles as a result of localized thermal fluctuations induced by the magnetic nanoparticles heated by the AC magnetic field and magnetic dipole-dipole interactions. The thermal fluctuations lead to an increase in the frequency of particle collisions and then their sticking together due to magnetic dipole-dipole interactions. It should be added that during this heating induced by the RFMF, the temperature of the top layer of the colloidal suspension, placed in an Eppendorf tube (see diagram (a) in Figure 1), goes through characteristic changes over time. Namely, the monotonic rise in temperature is accompanied by rapid temperature jumps caused by the convective upward flow of warm liquid from hot aggregates that previously fell to the bottom by gravity. The question is to what extent can we counteract these unwanted agglomeration processes?

A common way to break up aggregates and promote nanoparticle dispersion is ultrasonic technology [18]. Therefore, an effective way of delivering acoustic energy is key to sonicating the colloidal suspension. In our case, the ultrasonication process was done by a probe-type ultrasonic homogenizer. To avoid the temperature rise of the suspensions due to the applied high-energy ultrasonic waves, the sample vials were placed in an ice-water bath (see diagram (b) in Figure 1) so that the temperature of the ultrasonic suspension during the sonication process never exceeds 20 °C. It is usually believed that the zeta potential value is considered to increase for nanoparticles exposed to high-frequency acoustic waves [19,20], and it is generally accepted that as sonication time and power increase, the average cluster size decreases and the stability of the nanoparticle suspension improves, although this process has not yet been thoroughly investigated. At the same time, as numerous examples have shown, when dealing with very high sonication powers and long sonication times, dispersing agglomerates can be inhibited or even partially reversed [19,21,22].

Our research aims to highlight the role of the undesirable agglomeration processes taking place in a suspension of magnetic nanoparticles subjected to an AC magnetic field, which may result in a reduction in the amount of heat generated. Since ultrasonic treatment is considered to be the primary means of dispersing aggregated particles, we investigate its effect on the efficiency of magnetic heating, including the importance of selecting the optimal treatment time. At the same time, based on detailed measurements using the DLS technique, we can observe a power-like trend for the decreasing average hydrodynamic diameter of magnetic nanoparticles with increasing sonication time, which, to our knowledge, has not been studied before.

## 2. Materials and Methods

### 2.1. Synthesis

The synthesis of the iron oxide nanoparticles was carried out according to the following procedure. The amount of 3.81 g of FeSO_4_·7H_2_O was dissolved in 100 mL of 0.02 M hydrochloric solution. The 7.41 g of FeCl_3_·6H_2_O was then dissolved in 100 mL of distilled water. Solutions were mixed in one beaker and stirred at 400 rpm for 50 min in a nitrogen atmosphere. After this time, the rotational speed of the mechanical stirrer was increased to 800 rpm and 25 mL of a 25% ammonia solution was added dropwise at a rate of 1 drop/sec. The resulting suspension was mixed for 30 min at a speed of 800 rpm. To improve the colloidal stability, the appropriate amount of the magnetic nanoparticle suspension was then taken and subjected to a sodium base washing procedure until the pH was 11.5. A neodymium magnet was used to separate particles from the suspension during the washing process. The obtained suspension was stored in a base solution.

### 2.2. Measurement Characterization

Several measurement techniques were used to characterize the properties of the synthesized nanoparticles:Transmission electron microscopy (TEM) with the energy dispersive spectrometer (EDS) to determine the size of nanoparticles and confirm their chemical composition,Raman spectroscopy in a backscattering configuration on Renishaw InVia Quantor spectrometer (Renishaw, Stone, UK) (using 532 nm diode laser) to identify the chemical composition of the synthesized material,the dynamic light scattering (DLS) method on the Zetasizer Nano ZS (Malvern Panalytical, Malvern, UK) to characterize the nanoparticles suspension in terms of the particle size distribution,Non-contact measurements with an infrared thermometer on the Optris CTlaser LT-CF1 (Optris GmbH, Berlin, Germany) model equipped with a double laser sight with an optical head 75:1 or 50:1 with a spectral range from 8 to 14 µm,Sonics VCX 130 ultrasonic homogenizer (Sonisc & Materials, Inc., Newtown, USA) (operating at a frequency of 20 kHz with a net power output of 130 W and the amplitude set to 100%) to disperse iron oxide aggregates,The RFMF generator (operating at a frequency of 100 kHz and 20 mT amplitude) for the radio frequency magnetic heating (RFMH).

The transmission electron microscopy (TEM) technique is one of the most effective analytical tools that can give direct information on the morphological and structural properties of MNP. Figure 2 presents nanoparticle assemblies from a dried magnetic colloidal suspension in an alkaline solution with a pH value of 11.5. The TEM image shows that the nanoparticles are polydisperse and more or less spherical. Moreover, they indicate that the size distribution can be well approximated by a log-normal function, where the average size of the nanoparticles is about 11.7 nm with a spread of σ = 0.24.

In practice, bare magnetite (Fe_3_O_4_) nanoparticles are highly chemically active and can easily oxidize in air or water to form maghemite (Fe_2_O_3_). Therefore, a small fraction of maghemite is present. For our MNP, this scenario has been confirmed by the Raman spectrum presented in panel (c) in in Figure 2b, where the most characteristic magnetite peak dominates near 670 1/cm, but is accompanied by a smaller maghemite peak centered around 710 1/cm [23]. To check whether the chemical composition of nanoparticles changes under the influence of sonication, Raman measurements were performed for all representative samples, which showed that neither mechanical waves, such as ultrasound, nor the RFMH process has a significant influence on the relative ratio of magnetite and maghemite.

Since DLS is a proven technique for assessing the particle size, size distribution, and zeta potential of nanomaterials in solution, in each of these cases, a sample of the suspension was taken and tested in the Zetasizer Nano ZS after dilution. The most common DLS software allows the expression of the particle size distribution in terms of intensity, number, and volume, but this usually leads to three different estimates of the hydrodynamic diameter (HD). The HD is the diameter of a fictitious hard sphere that diffuses at the same speed as the actual particles studied in DLS. As in fact the operating principle of the DLS is based on the measurement of the scattered light intensity, the representative HD value should be derived from the intensity measurements, while the other two parameters (volume and number) can be used as supporting information.

The magnetic heating experiments were performed using a laboratory-made experimental setup for magnetic heating. This device uses the RFMF generator, which is based on a coil made from a copper tube, which has six turns. The water cooling system is applied, where water flows through the coil interior (see diagram (a) in Figure 1). Before starting the measurements, each sample was cooled down to the cooling water temperature.

### 2.3. Magnetic Heating

The specific absorption rate (SAR) is a measure of the ability of MNP to generate heat, which is defined as the heat generated per unit time by a unit mass of MNP. It has been confirmed that the SAR is governed by various physical and magnetic properties of the nanoparticles [24]. The RFMF supplies energy to the nanoparticle system through two dominant mechanisms, namely the Néel relaxation and the Brownian relaxation. When the thermal energy is dissipated by the rearrangement of atomic dipole moments within the crystal and the particle moment relaxes to its equilibrium orientation, there is so-called Néel relaxation:(1)$$\tau_{N}=\tau_{0}\exp\left(KV/k_{B}T\right),$$
where $\tau_{0}$ is a time constant that is generally of 10^−9^ s, *K* is the anisotropy constant, and *V* is the primary volume of MNP. Another loss type may arise in the case of ferrofluids, which is related to the rotational Brownian motion of the magnetic particles. In this case, the heat energy is transferred to the liquid by shear stress:(2)$$\tau_{B}=3\eta V_{H}/k_{B}T,$$
where η is the medium viscosity and $V_{H}$ is the hydrodynamic volume. When the properties of nanoparticles objects, ranging in size from 1 nm to 100 nm, are considered, the role of Brownian relaxation is dominant in the case of larger nanoparticles, while the importance of the Néel processes increases significantly and begins to prevail for smaller particles [3,25].

Magnetic anisotropy may come from two distinct sources: crystalline anisotropy and shape anisotropy. The former is a property of the crystalline structure itself. Since the magnetic moment has a preferred alignment direction along a crystalline axis, any deviation from this direction comes at the expense of magnetic free energy. Each magnetic nanoparticle exhibits an anisotropy, the presence of which leads to a minimum of free energy along a specific axis. Magnetic anisotropy free energy refers to the free energy barrier associated with the rotation of the magnetic moment of the nanoparticle away from this preferred axis. This energy barrier makes Brownian rotation possible—if the barrier is large enough, the entire particle can be rotated against viscous drag by an applied magnetic field. However, the role of anisotropy is particularly significant in the Néel relaxation, due to the presence of the exponential term (see Equation (Equation 1)), which then translates into its significant impact on SAR [26].

## 3. Results and Discussion

To evaluate the stability of the MNP suspension, its behavior was observed for different storage times and different values of pH ranging from 7 to 13 [17]. Based on these studies, we found that colloidal suspensions of bare nanoparticles appeared to be the most stable at pH = 11.5. Therefore, when we analyzed the effect of magnetic hyperthermia on the destabilization of magnetic colloidal suspensions, this case was used as a reference point. For iron oxide nanoparticles synthesized for the present study, the zeta potential value for the suspension dispersed in an alkaline solution at pH = 11.5 takes the value of about −31.7±1.7 mV, which implies a strong electrostatic repulsion between the aggregates.

To check the sonication effect, a volume of 10 mL of the same MNP suspension was subjected to an ultrasonic homogenizer. After a series of sonication processes of varying duration, a volume of 2 mL of the suspension for each case was placed in an Eppendorf tube and its temperature was equalized with the temperature of water $T_{water}^{in}$ entering the coil (typically around 17 °C). As shown in panel (a) in Figure 3, even a 1-min sonication leads to the breakdown of numerous aggregates of nanoparticles and thus to a shift of the maximum of the intensity-based size (the HD) distribution of dispersed particles. Subsequently, longer and longer sonications deepen this trend, as evidenced by the shift of successive distribution function maxima toward smaller and smaller nanoparticle sizes. Recently, similar results for iron oxide nanoparticles placed in deionized water have shown that their average size decreases with increasing homogeneity of the solution after ultrasonic treatment [27]. Our experimental data suggest that the functional dependence of the maxima positions on the sonication time is of a power nature. Moreover, the positions of the peaks, and thus the HD values, converge to a final value (the offset) as the sonication time increases to infinity. This is further supported by the fact that the offset value is in acceptable agreement with the value of the average size of a single nanoparticle determined by the TEM technique (see the dash-dotted line in panel (a) in Figure 3). The volume-based and number-based distributions are collected in the Appendix A. In both cases, the power trend of decreasing average hydrodynamic diameter of magnetic nanoparticles with increasing sonication time is suggested, although the values of exponents and offsets are slightly different.

The values of the zeta potential—estimated from experimentally determined electrophoretic mobilities of agglomerates—are presented in panel (b) in Figure 3. They suggest the increase of the zeta potential with the sonication time. However, it should be noted that the used particle number and zeta potential analyzer gives measurement results based on the Smoluchowski model. The observed measurement result could also be influenced by the magnetic dipole-dipole interactions between magnetic nanoparticles. Thus, in Figure 3b we observe an increase in the zeta potential after sonication, but a more complex model, taking into account the specificity of the magnetic suspension, should probably be used. A separate issue is that of re-agglomeration of the iron oxide nanoparticles, because the more dispersed the nanoparticles, the greater the excess free energy associated with the presence of the surface. As a result, after some time of continuous sonication, the number of nanoparticles undergoing re-agglomeration becomes comparable with the number of particles formed during the collisions of agglomerates. A similar effect has recently been observed for non-magnetic nanofluids, showing that there is an optimal sonication time to obtain the best dispersion of the suspension in the ultrasonic process [20].

In order to test the ability of the MNP suspension to release heat, it was subjected to an external RFMF, during which ΔT was monitored (see Figure 1). Most studies on magnetic heating to estimate the power generated by MNP assume that the heat loss is linearly related to the temperature difference between the sample and its surroundings. The linear increase of the ΔT function at the initial stage of magnetic heating translates into the possibility of determining the SAR coefficient, the value of which is proportional to the derivative of the temperature difference with respect to time (called the “slope” in Figure 4). Measurements show that as the pre-sonication time increases, a significant increase in the SAR value of the MNP suspensions is observed (see Appendix A). Moreover, as the sonication time increases, as shown in Figure 4, the final elevation temperature ($T_{\mathrm{elev}}=\Delta T$ (t = 30 min)) also increases. Overall, this effect of extending the sonication time can be explained by the fact that the longer the ultrasonic acoustic energy is delivered, the more single domain nanoparticles appear in the suspension. Their average size of about 12 nm makes them primarily heated by the Néel relaxation processes. Moreover, in the case of finely divided material, when more and more agglomerates consist of only a few single-domain nanoparticles, the AC magnetic field favors the correlation of the orientation of the magnetic moments of individual nanoparticles in such an aggregate, leading, on average, to their parallel alignment. This increases the effective volume of such an agglomerate, which can be interpreted as a single-domain nanoparticle. As the Néel relaxation time then also increases (see Equation (Equation 1)), it causes a delay in the relaxation of the magnetic moment in relation to the oscillation of the AC magnetic field and thus induces additional heat release. We expect such an effect only in the case of agglomerates, which are clusters of only a few single-domain nanoparticles, because, in the case of larger agglomerates, too many individual magnetic moments cannot be correlated under the influence of the external AC magnetic field.

When analyzing the shape of the ΔT curves for unsonicated and sonicated samples, it can be noticed that for the former, there was a jump after some time of exposure to the RFMF, as described in our previous article [17]. For our samples, as the plots in Figure 4 demonstrate, such a jump can be observed after about 5 min. This behavior is related to the agglomeration processes of clusters in the suspension during the RFMH process and sedimentation of large aggregates under gravity in the vial. The bottom layer is heated by the warm aggregates of magnetic nanoparticles, which in turn causes the convective upward flow of warm fluid from the bottom layer. When the aggregates descend to the bottom, they heat the liquid present there and can be picked up, along with the rising warm liquid, participate in subsequent collisions, and fall to the bottom again. Such a circulation of the liquid together with the fine particles, results in a comb-like shape of the ΔT function. In the case of suspensions subjected to ultrasound treatment for more than a minute, the average aggregate size is reduced notably, therefore the influence of agglomeration processes on the shape of the ΔT curves is greatly limited. The result is a smooth and monotonic course of the ΔT function, as presented in Figure 4.

As shown in panel (b) in Figure 4, although the procedure for preparation of the suspension, sonication, and magnetic heating for each sample was the same, individual measurements of the ΔT function may yield slightly different results. For the samples not subjected to sonication, the slope values of the ΔT curves are estimated to be 2.16 °C/min within a range of 0.48 °C/min (22.3%). After 15 min of sonication, the estimated value is 4.26 °C/min within a range of 0.45 °C/min (10.5%), whereas, after 120 min, the estimated value is 5.0 °C/min within a range of 0.4 °C/min (7.9%). As far as T_elev_ is considered, for the samples not subjected to sonication, T_elev_ is estimated to be 24.79 °C/min within a range of 1.53 °C/min (6.2%). For samples after 15 min of sonication, T_elev_ is estimated to be 35.54 °C/min within a range of 1.48 °C/min (4.2%), while, after 120 min, the estimated T_elev_ is 39.10 °C/min within a range of 0.51 °C/min (1.3%). Thus, the longer the sonication time, the more homogeneous the colloidal suspension becomes, and thus we observe smaller spreads for both slope values and elevation temperatures. Taking into account the results from Figure 4, it can be seen that an increase of sonication time of the magnetic nanoparticle suspension results in a significant increase in its heating efficiency. However, above a certain time, further sonication seems unjustified, because then the inevitable fluctuations of the elevation temperature for individual RFMH processes become comparable with the already slight increase in the average value of the elevation temperature.

At this point, let us answer the question of to what extent magnetic heating affects the size distribution of suspensions previously subjected to the sonication process of different duration. Figure 5 presents the intensity-based size distributions, before and after magnetic heating, for a few exemplary sonications. The case of untreated suspension, the starting point for this study, was discussed in our previous article [17]. We were then interested in the efficiency of filtering large agglomerates from the suspension, so our analysis focused on the particles remaining in the supernatant. In the current study, the magnetically heated suspension is shaken first before DLS measurement, so there is no loss of nanoparticles, and an overall shift in the size distribution towards the larger size is observed. Comparing each top panel with the corresponding bottom panel in Figure 5, it can be seen that, in general, the magnetic heating causes an increase in the number of aggregates in all tested suspensions. For suspensions that have been sonicated for half a minute and not sonicated, the size distributions are significantly shifted towards the larger value. On the other hand, for suspensions that are sonicated for a long time, the changes are no longer as pronounced, but an increase in the number of agglomerates is also noticeable.

Our results also allow us to analyze changes in the NMP suspension over time. Namely, for a suspension not subjected to sonication or RFMH, the size distribution becomes more uniform over time (compare top panels in Figure 5). On the other hand, for a briefly sonicated (half a minute) suspension, the average size of the aggregates is noticeably reduced compared to the non-sonicated suspension, but the passage of time alone does not change the size distribution. However, in the case of the longest sonicated suspension, where the average size of aggregates is significantly reduced, the passage of time leads to re-agglomeration and then to a certain shift in the size distribution towards large agglomerates.

## 4. Conclusions

Recently, it has been shown that a magnetic colloidal suspension can be destabilized by exposure to the RFMF [17]. The resulting agglomeration process of magnetic nanoparticles may lead to undesirable phenomena, such as weakening the effect of hyperthermia on colloidal suspensions of magnetic nanoparticles. Therefore, the purpose of this article was to determine the optimal conditions for the formation of a thermodynamically stable mixture resistant to the aggregation process during magnetic hyperthermia.

In the present study, particular attention was paid to the role of sonication in the preliminary preparation of magnetic colloids. A series of measurements were required to evaluate changes in the size distribution of suspended particles for various sonication times. The size distribution of dispersed nanoparticles and the zeta potential were analyzed by dynamic light scattering. It is worth noting that a power-like trend of decreasing values of the hydrodynamic diameter with increasing sonication time was observed. This functional dependence holds for each of the distributions based on intensity, volume, or number weighting, although the values of exponents and offsets are slightly different. More research is certainly needed to obtain more conclusive evidence.

When the magnetic heating for nanoparticles was considered, an overall shift of the size distribution towards the larger size was observed. By repeating the RFMH process several times for samples prepared in the same way, we were able to assess the reliability of the measurement results, e.g., for the SAR coefficient, and found that increasing the sonication time of the MNP suspension results in a noticeable increase in its heating efficiency. Nevertheless, in practice, extending the sonication time beyond 15 min does not lead to a further increase in heating efficiency.

## Figures and Tables

**Figure 1 nanomaterials-12-03064-f001:**
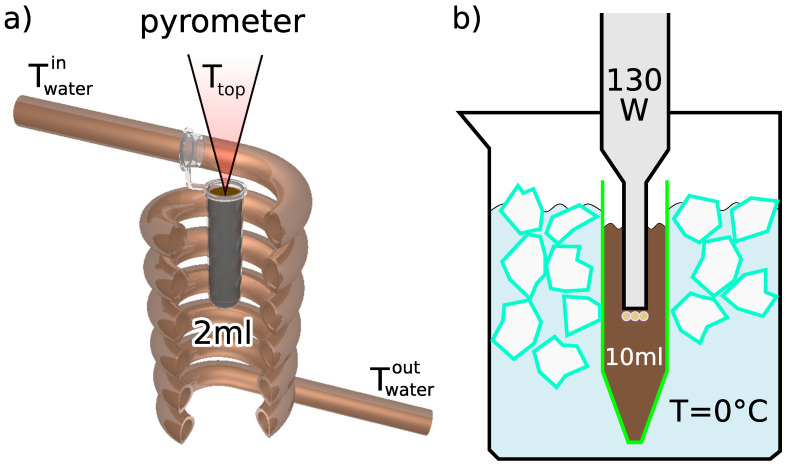
(**a**) The schematic drawing of the induction heating set, where the temperature is different $\Delta T=T_{top}-T_{water}^{in}$ between the top layer of the aqueous solution (measured with a pyrometer) and the water entering the cooling system. (**b**) Diagram of our ultrasonic probe placed in a vial with a colloidal suspension. Thanks to the ice-water bath, the temperature of the sonicated suspension never exceeds 20 °C during the sonication process. After transferring the volume of 2 mL of suspension from the vial to the Eppendorf tube, its temperature was equalized with the temperature of water $T_{water}^{in}$ entering the coil (typically around 17 °C).

**Figure 2 nanomaterials-12-03064-f002:**
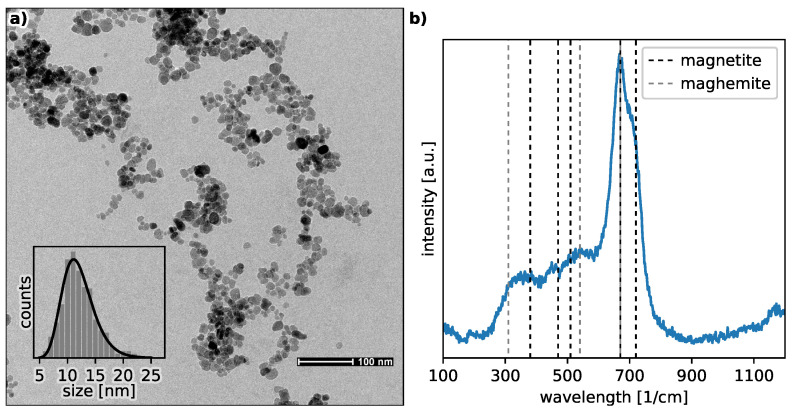
Panel (**a**) shows the TEM image for the bare MNP from the solution at the pH=11.5. The nanoparticle-size distribution histogram and the log-normal function with the fitted parameters μ = 11.7 nm and σ = 0.24 are presented as the inset. Panel (**b**) presents the Raman spectrum for the bare MNP with the positions of distinctive Raman peaks for magnetite and maghemite.

**Figure 3 nanomaterials-12-03064-f003:**
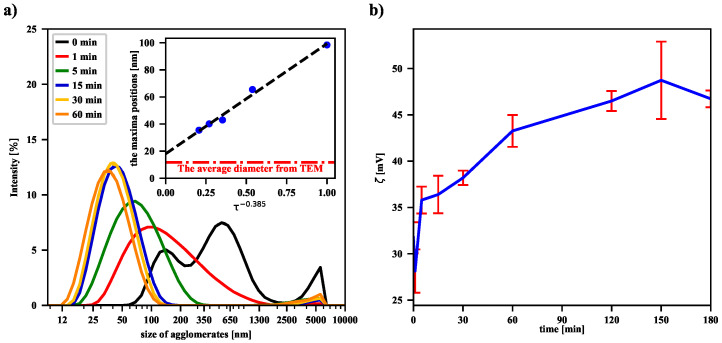
(**a**) The intensity-based size distribution of the nanoparticle agglomerates dispersed in an aqueous suspension with pH = 11.5 for different sonication durations. The inset suggests the power-law dependence of the mean HD on the duration of the sonication process, where the HD offset is 17.6 nm. The dimensionless quantity τ is defined as (the sonication time)/(1 min). (**b**) The dependence of the zeta potential on the sonication duration.

**Figure 4 nanomaterials-12-03064-f004:**
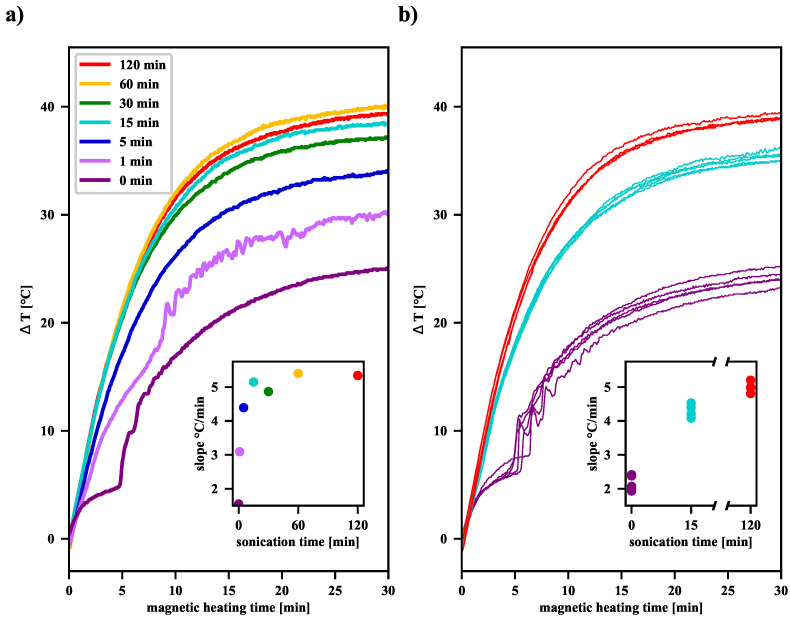
Both panels show how ΔT changes during the RFMH process for suspensions treated with ultrasound. Before the sample was placed in the coil of the RFMF generator, so its temperature was equalized with the temperature of the water entering the cooling system; therefore, the starting ΔT was around zero. Panel (**a**) shows the dependency of ΔT on the time of magnetic heating for suspensions treated with ultrasound. The series of curves on the panel (**b**) presents the spread of the ΔT curves for the three series of measurements, where each series of samples was taken from the same vial. Each series of samples was taken from the same vial (with the suspension not sonicated or sonicated for 15 or 120 min, respectively). In both insets, the initial slopes of the ΔT curves were determined during magnetic heating in the range of 30–90 s when the ΔT increase rate was almost linear. Measurements on the two panels were taken on different days, but all samples were from the same synthesis.

**Figure 5 nanomaterials-12-03064-f005:**
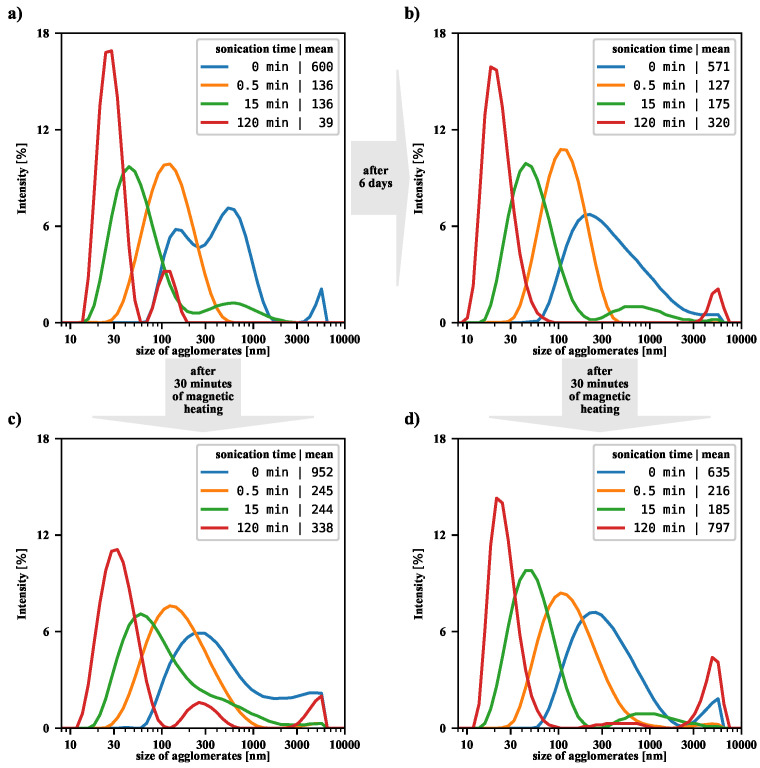
The graph shows the intensity-based size distributions for a suspension initially subjected to different durations of ultrasonic treatment. (**a**,**b**): without the additional RFMH; (**c**,**d**): additionally subjected to the 30’ RFMH (performed the day before DLS measurements); (**a**,**c**): measurements were taken 1 day after nanoparticle synthesis; (**b**,**d**): measurements were taken 7 days after the synthesis of nanoparticles. For each panel in the last column of the legend, the value of the mean size distribution for each case is included. The graph shows the intensity-based size distributions for suspension pre-treated with ultrasound. The top left panel shows DLS results for the colloidal suspension of nanoparticles made the previous day (without using RFMH). For each drawing, the last column of the legend contains the mean size distribution for each case.

## Data Availability

Not applicable.

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
