# Peer review of "Effect of Magnetic Heating on Stability of Magnetic Colloids"

_nanomaterials, 2022, doi:10.3390/nano12173064_

Round 1
Reviewer 1 Report
I read the paper with interest, considering the significance of the topic. I think that it is a good contribution to Nanomaterials journal, although some points should be considered by the authors:
1. Line 103: the particles are set at pH 11.5. Perhaps it would be good to specify that this is done in order to maximize their zeta potential and stability. However, it must be also emphasized that these are not conditions applicable in real hyperthermia.
2. Line 128: since a log-normal distribution fits the size data, the spread of the distribution, not just the average, should be provided
3. The increase in zeta potential with sonication in Fig. 3 is not justified, unless it reflects the increase in mobility and the zeta potential must be then evaluated using a model not as simple as Smoluchowski's
4. A small point: the color display in Fig. 4a should be speified in order (now reads 0,1,5, 30, 15,120, 60 min; this is confusing for the reader)
5. The data in Fig. 5 should be accompanied by a quantitative Table: indicate at least the averages before and after magnetic treatment with their uncertaintities and statistical tests of differences. This would confirm the effect of magnetic heating
Author Response
2022-08-20 Dear Editor, thank you for sending us referees' reports on our manuscripts nanomaterials-1844430by A. Drzewinski, M. Marc, W.W. Wolak, M.R. Dudek. We thank the referees for a careful and critical reading of our manuscript and for the positive judgment. In the revised manuscript we have addressed all points that have been raised. To make it easier to trace the applied changes in the latex file, the green color of the font indicates the earlier form of the text (in the final version, these passages should be removed),
while the red font indicates the revised version. Yours faithfully Andrzej Drzewiński on behalf of the authors Details of changes Review report 1 1) Line 103: the particles are set at pH 11.5. Perhaps it would be good to specify that this is done in order to maximize their zeta potential and stability. However, it must be also emphasized
that these are not conditions applicable in real hyperthermia. As suggested by the reviewer, the sentence "Then, the appropriate amount of the suspension was taken and the washing procedure was carried out with a sodium base solution of the pH value equal to 11.5." has been changed to "Then, to improve the colloidal stability, the appropriate amount of the magnetic nanoparticle suspension was taken and subjected to a sodium base washing procedure until the pH was 11.5." Moreover, the sentence "Of course, taking into account medical applications, such a pH value is far from normal physiological conditions." has been added. 2) Line 128: since a log-normal distribution fits the size data, the spread of the distribution, not just the average, should be provided The sentence was supplemented with the spread of the distribution "Moreover, they indicate that the size distribution can be well approximated by a log-normal function, where the average size of the nanoparticles is about 11.7 nm with a spread of σ = 0.24." 3) The increase in zeta potential with sonication in Fig. 3 is not justified, unless it reflects the increase in mobility and the zeta potential must be then evaluated using a model not as simple
as Smoluchowski's As suggested by the reviewer, the sentence "Moreover, our measurements of the zeta potential for nanoparticles after sonication clearly show that its value increases with increasing sonication duration." has been reformulated "The values of the zeta potential - estimated from experimentally determined electrophoretic mobilities of agglomerates – are presented in the panel b) in Fig.3. They suggest the increase of
the zeta potential with the sonication time. It should be noted, however, that the used particle number and zeta potential analyzer gives measurement results based on the Smoluchowski model.
The observed measurement result could also be influenced by the magnetic dipole-dipole interactions between magnetic nanoparticles. Thus, in Fig.3b we observe an increase in
the zeta potential after sonication, but probably a more complex model, taking into account the specificity of the magnetic suspension, should be used." 4) A small point: the color display in Fig. 4a should be speified in order (now reads 0,1,5, 30, 15,120, 60 min; this is confusing for the reader) In accordance with the reviewer's comments, Figure 4A has been changed. 5) The data in Fig. 5 should be accompanied by a quantitative Table: indicate at least the averages before and after magnetic treatment with their uncertaintities and statistical tests of differences.
This would confirm the effect of magnetic heating To support our conclusions with quantitative data, Figure 5 has been revised. A numerical value for the mean particle size distribution has now been placed on each panel, so that by
comparing the corresponding values before and after the magnetic heating process with each other, one can assess its effect on the particle size distribution.
Reviewer 2 Report
Referee’s report on the manuscript:
nanomaterials-1844430“Effect of magnetic heating on stability of magnetic colloids”
by Andrzej Drzewinski et al.
The manuscript entitled “Effect of magnetic heating on stability of magnetic colloids” by Andrzej Drzewinski et al. describes the stability of magnetic colloids after the application of a magnetic field.
Overall, the effect on magnetic colloid subjected to a magnetic field to produce heating and of course the magnetic hyperthermia therapy is of great current interest with many groups working in this area. The work is according to the topic of the journal. My RECOMMENDATION is ACCEPTABLE for publication with MAJOR REVISIONS, if the author makes some corrections to improve the manuscript as indicated.
· Language of the text could be improved to make it more fluent and the many misprints on the manuscript must be corrected before publishing
· Page 2 lines 89 and following. The authors organize the manuscript in the form of chapters, but it is not clear to me, as it is not a book, and they do not explicitly mention the chapters, but only the title. This should be corrected, and this chapter organization should be removed, as it is not present.
· - Figure 1 shows the schematic drawing of the induction heating assembly, where a 2 mL Eppendorf can be seen while on the right side, the sonication is in a 10 mL Eppendorf. The author takes from these 10 mL, 2 mL of nanoparticles at 0ºC. But when they take this volume, how can they be sure that the temperature is still 0°C?
· How can the author be sure that this temperature of 0°C will be maintained during the whole experiment, since it lasts more than 20 min. And it is not explained whether there is a thermal chamber around the system to make it adiabatic. If the temperature is not kept constant, the Eppendorf will gradually increase in temperature, up to the ambient temperature and can be confused with the temperature caused by the magnetic field.
· Starting from such a low temperature (0ºC) without any insulation, it is very difficult to obtain the SAR, as the system will dissipate energy by exchanging it with the room temperature.
· The temperature of the total system is not specified.
· In relation to the previous question, the temperature of the water in the induction coil must be specified, which must be below 0ºC. This is very difficult if water is used. Therefore, a different coolant must be used. Furthermore, it is necessary to maintain this temperature throughout the experiment when applying current to the coil, which will surely increase its temperature due to the Joule effect.
· How the author measures the magnetic field strength of 20mT in the coil of the RFMF generator. To be explained or measured.
· How the author measures the temperature with a pyrometer, no details of the pyrometer are given. The brand, model, characteristics, ranges, etc. should be written.
· The temperature of a sample without the application of a magnetic field must be measured to ensure the adiabatic condition and that the temperature rise is only due to the influence of the magnetic field.
· Figure 4 shows the temperature changes. The author wrote that in both starts 30-90 m, the temperatute is almost linear, but in figure 4b, inset appears to be exponential, and in F4B, it is linear, as shown respectively. Is this so?
· How do you explain the two slopes after 5 minutes for the 0 sonication samples in figures a and b?
· On page 5 line 11 and following the SAR coefficient is explained, but in the result the author has not calculated it. This value should be placed in a table, to underline and characterize this property of the sample as a nano-heating element.
· The SAR value, if calculated, or in such a case, the values quoted do not seem to coincide with the increase in slope temperature. As we can see in Fig4. For example the slope of the green curve of 30 m sonication in 5 minutes, the temperature increases almost 20 degrees, if this is true this value is 4 degrees/min and not 0.481 º C/min. Also, for the blue curve of 5 m of sonication the temperature increases by about 15 °C and 5 min, right? This should be explained in detail
· SAR value must be calculated of each sample.
· Page 10 line 332. Watt means this text “However, it shoud be noted that extending the sonication time beyond dosen minutes. What the meaning “dosen”?
Author Response
2022-08-20 Dear Editor, thank you for sending us referees' reports on our manuscripts nanomaterials-1844430by A. Drzewinski, M. Marc, W.W. Wolak, M.R. Dudek. We thank the referees for a careful and critical reading of our manuscript and for the positive judgment. In the revised manuscript we have addressed
all points that have been raised. To make it easier to trace the applied changes in the latex file, the green color of the font indicates the earlier form of the text (in the final version,
these passages should be removed), while the red font indicates the revised version. Yours faithfully Andrzej Drzewiński on behalf of the authors Details of changes Review report 2 1) Language of the text could be improved to make it more fluent and the many misprints on the manuscript must be corrected before publishing The manuscript has been carefully revised to improve the grammar and readability. 2) Page 2 lines 89 and following. The authors organize the manuscript in the form of chapters, but it is not clear to me, as it is not a book, and
they do not explicitly mention the chapters, but only the title. This should be corrected, and this chapter organization should be removed, as it is not present. As suggested by the reviewer, the chapter organization has been removed 3) Figure 1 shows the schematic drawing of the induction heating assembly, where a 2 mL Eppendorf can be seen while on the right side, the sonication is
in a 10 mL Eppendorf. The author takes from these 10 mL, 2 mL of nanoparticles at 0ºC. But when they take this volume, how can they be sure that
the temperature is still 0°C? As our description of Figure 1 was not clear, the figure caption was expanded from "b) Diagram of our ultrasonic probe surrounded by an ice-water bath." to "b) Diagram of our ultrasonic probe placed in a vial with a colloidal suspension. Thanks to the ice water bath, the temperature of the sonicated suspension
did not exceed twenty-some degrees Celsius during the sonication process. After transferring the volume of 2 ml of suspension from the vial to the Eppendorf
tube, its temperature was equalized with the temperature Twaterin of the water entering the coil (typically around 17ºC)." 4-7) - How can the author be sure that this temperature of 0°C will be maintained during the whole experiment, since it lasts more than 20 min. And it is not explained
whether there is a thermal chamber around the system to make it adiabatic. If the temperature is not kept constant, the Eppendorf will gradually increase in
temperature, up to the ambient temperature and can be confused with the temperature caused by the magnetic field. - Starting from such a low temperature (0ºC) without any insulation, it is very difficult to obtain the SAR, as the system will dissipate energy by exchanging
it with the room temperature. - The temperature of the total system is not specified. - In relation to the previous question, the temperature of the water in the induction coil must be specified, which must be below 0ºC. This is very difficult if water
is used. Therefore, a different coolant must be used. Furthermore, it is necessary to maintain this temperature throughout the experiment when applying current
to the coil, which will surely increase its temperature due to the Joule effect. We believe that the changes in the caption of Figure 1, along with the other comments in the text, clearly refers to the doubts raised in these points. 8) How the author measures the magnetic field strength of 20mT in the coil of the RFMF generator. To be explained or measured. The magnetic field strength in the coil is estimated with the LA 100-P Current Transducer by LEM International SA. The device performs a non-contact
measurement of the current fed to the coil based on the Hall effect. From the electric current, the strength of a homogeneous magnetic field in the measuring area
inside the coil is calculated. 9) How the author measures the temperature with a pyrometer, no details of the pyrometer are given. The brand, model, characteristics, ranges, etc. should be written. The subsection "Measurement characteristics" has been supplemented with information on the measuring instrument for non-contact temperature measurement. "non-contact measurements with an infrared thermometer on the Optris CTlaser LT-CF1 model equipped with a double laser sight with an optical head 75:1 or 50:1 with
a spectral range from 8 to 14 μm". 10) The temperature of a sample without the application of a magnetic field must be measured to ensure the adiabatic condition and that the temperature rise is only due
to the influence of the magnetic field. Before measuring the temperature of the suspension in the presence of an alternating magnetic field, we verified that the temperature of the suspension in an
Eppendorf tube in the absence of the field did not change. As mentioned earlier, the temperature of the suspension in an Eppendorf tube was always pre-equilibrated
with the temperature of the water entering the cooling system. 11) Figure 4 shows the temperature changes. The author wrote that in both starts 30-90 m, the temperatute is almost linear, but in figure 4b, inset appears to be
exponential, and in F4B, it is linear, as shown respectively. Is this so? As the measured quantity "the temperature difference" shown in Figure 4 was not clearly defined enough, several changes were made. In order to clearly give
the definition of ΔT, the caption of Figure 1 (where ΔT appears for the first time) has been changed from "a) The schematic drawing of the induction heating set, where the temperature difference between the top layer of the aqueous solution (measured with a pyrometer)
and the water entering the cooling system is monitored." to "a) The schematic drawing of the induction heating set, where the temperature difference ΔT = Ttop - Twaterin between the top layer of the aqueous solution (measured
with a pyrometer) and the water entering the cooling system is monitored." Additionally, the symbol Ttop has been included in Figure 1. The passage "In order to test the ability to release heat by the MNP suspension, it was subjected to an external RFMF (see Fig.4) and the temperature difference (ΔT) between
the top layer of the aqueous solution and water entering the cooling system was monitored (see the diagram (b) in Fig.1)." has also been reworded to avoid repetition "In order to test the ability to release heat by the MNP suspension, it was subjected to an external RFMF (see Fig.4) and ΔT was monitored (see Fig.1)." Moreover, on the ordinates of both graphs in Figure 4, the designation "temperature difference" has been replaced with the symbol ΔT. At the same time,
the following sentence was added to the caption of this figure: "Because before the sample is placed in the coil of the RFMF generator, its temperature is equalized with the temperature of the water entering the cooling system,
the starting ΔT is around zero." 12) How do you explain the two slopes after 5 minutes for the 0 sonication samples in figures a and b? In order to properly address the temperature jumps in the ΔT plot for a suspension of nanoparticles not subjected to sonication, as discussed in detail in
our previous paper in Nanomaterials, we have changed the passage "When analyzing the shape of the ΔT curves for unsonicated and sonicated samples, it can be noticed that for the former there is a jump after a certain time of
exposure to RFMF superimposed on its monotonic increase. This behavior is related to the agglomeration processes of clusters in the suspension and sedimentation
of large aggregates under gravity in the vial." to the following form "When analyzing the shape of the ΔT curves for unsonicated and sonicated samples, it can be noticed that for the former, there is a jump after some time of exposure
to the RFMF as described in our previous article [17]. For our samples, as the plots in Fig.4 demonstrate, such a jump can be observed after about 5 minutes. This
behavior is related to the agglomeration processes of clusters in the suspension during the RFMH process and sedimentation of large aggregates under gravity in the vial." 13, 15) - On page 5 line 11 and following the SAR coefficient is explained, but in the result the author has not calculated it. This value should be placed in a table, to underline
and characterize this property of the sample as a nano-heating element. - SAR value must be calculated of each sample. As suggested by the reviewer, the SAR values were calculated for each case. The results in the form of tables were included in the Supplementary materials. Moreover, the sentence in the last section „Our results, including analysis of random deviations for Telev obtained by repeating a given RFMH process, allow us to conclude that the extension of the sonication time
of the MNP suspension causes a significant increase in its heating efficiency.” has been changed „By repeating the RFMH process several times for samples prepared in the same way, we were able to assess the reliability of the measurement results, e.g. for the SAR
coefficient, and found that increasing the sonication time of the MNP suspension results in a noticeable increase in its heating efficiency.” 14) The SAR value, if calculated, or in such a case, the values quoted do not seem to coincide with the increase in slope temperature. As we can see in Fig4. For example
the slope of the green curve of 30 m sonication in 5 minutes, the temperature increases almost 20 degrees, if this is true this value is 4 degrees/min and not 0.481 º C/min.
Also, for the blue curve of 5 m of sonication the temperature increases by about 15 °C and 5 min, right? This should be explained in detail Since the discussion of the graphs in Fig.4 was inaccurate and confusing, the following section "For the samples not subjected to sonication, the slope values of the ΔT curves are within a range of values with a spread of 0.481 ºC/min, which is 22.3% relative to the ΔT
mean value. After fifteen minute of sonication there is a spread of 0.448 ºC/min (10.5%) whereas after 120 minutes a spread of 0.396 ºC/min (7.9%). As far as Telev is considered,
for the samples not subjected to sonication, Telev's are within a range of values with a spread of 1.531 ºC, which is 6.2% relative to the Telev mean value. For samples after fifteen
minutes of sonication, Telev's are within a range of values with a spread of 1.478 ºC (4.2%), while after 120 minutes there is a spread of 0.512 ºC (1.3%)." was replaced to "For the samples not subjected to sonication, the slope values of the ΔT curves are estimated to be 2.16 ºC/min within a range of 0.48 ºC/min (22.3%). After fifteen minute
of sonication the estimated value is 4.26 ºC/min within a range of 0.45 ºC/min (10.5%), whereas after 120 minutes the estimated value is 5.0 ºC/min within a range of 0.4 ºC/min
(7.9%). As far as Telev is considered, for the samples not subjected to sonication, Telev is estimated to be 24.79 ºC within a range of 1.53 ºC (6.2%). For samples after fifteen
minutes of sonication, Telev is estimated to be 35.54 ºC within a range of 1.48 ºC (4.2%), while after 120 minutes the estimated Telev is 39.10 ºC within a range of 0.51 ºC (1.3%)." 16) Page 10 line 332. Watt means this text “However, it shoud be noted that extending the sonication time beyond dosen minutes. What the meaning “dosen”? The sentence containing the spelling error "However, it shoud be noted that extending the sonication time beyond dosen minutes or so no longer leads to a further increase in heating efficiency." has been changed to the following form "Nevertheless, in practice, extending the sonication time beyond a quarter of an hour does not lead to a further increase in heating efficiency."
Round 2
Reviewer 1 Report
The authors have considered my previous recommendations. I recommend publication.
Author Response
Dear Editor, thank you for sending us referees' reports on our resubmitted manuscripts nanomaterials-1844430by A. Drzewinski, M. Marc, W.W. Wolak, M.R. Dudek. Once again, we would like to thank both reviewers for their detailed and in-depth reports. In addition,
we would like to thank the second reviewer for the comments on the revised manuscript which enabled
us to prepare the final version. Details of the recent changes are listed below. To make it easier to trace the applied changes in the latex file, the green color of the font indicates
he earlier form of the text (in the final version, these passages should be removed), while the red font
indicates the revised version. Yours faithfully Andrzej Drzewiński on behalf of the authors
Reviewer 2 Report
2nd Referee’s report on the manuscript:
nanomaterials-1844430“Effect of magnetic heating on stability of magnetic colloids”
by Andrzej Drzewinski et al.
After reading the answers providing by the author, I am still recommended Acceptable for publication, if the author do some corrections to improve the manuscript. MINOR REVISIONS have been made as indicated.
· Figure Caption:
The author does not have to explain in the figure caption what we see in the graphs or in the figure/image. The figure caption should only have a short description and in the main text the author should mention the figure and explain whatever it is. Please change all figure captions to a short text and insert in the main text the explanation of each of them. and the figure caption in the manuscript write a small text
· The author wrote that many of the answers are in the supplementary file, but this is still in latex, and I could not read it in this format. This should be converted to pdf. I could not check those files, nor the tables mentioned in the text as S1 and S2. (SAR calculation)
· Some typos detected in the text on page 9 line 304 -307 where the range only has Cº and not ºC/min - To be corrected.
· Figure 5. the X-axis, wrote "the size of the agglomerates and should write only “size of the agglomerates”.

Author Response
Dear Editor, thank you for sending us referees' reports on our resubmitted manuscriptsnanomaterials-1844430 by A. Drzewinski, M. Marc, W.W. Wolak, M.R. Dudek. Once again, we would like to thank both reviewers for their detailed and in-depth reports.
In addition, we would like to thank the second reviewer for the comments on the revised
manuscript which enabled us to prepare the final version. Details of the recent changes are
listed below. To make it easier to trace the applied changes in the latex file, the green color of the font
indicates the earlier form of the text (in the final version, these passages should be removed),
while the red font indicates the revised version. Yours faithfully Andrzej Drzewiński on behalf of the authors Details of changes Review report 2 1) The author does not have to explain in the figure caption what we see in the graphs or in
the figure/image. The figure caption should only have a short description and in the main text
the author should mention the figure and explain whatever it is. Please change all figure captions
to a short text and insert in the main text the explanation of each of them. And the figure caption
in the manuscript write a small text In accordance with the reviewer's comments, we moved some of the text from the figure
captions to the main text and omitted some. In addition, a small font size was set in all figure captions. 2) The author wrote that many of the answers are in the supplementary file, but this is still in latex,
and I could not read it in this format. This should be converted to pdf. I could not check those files,
nor the tables mentioned in the text as S1 and S2. (SAR calculation) Previously, the supplementary file was submitted in latex format by mistake. Thank you for
pointing this out to me, so now we are sending the file in pdf format. 3) Some typos detected in the text on page 9 line 304 -307 where the range only has Cº and not ºC/min
- To be corrected. The listed typos have been corrected. 4) Figure 5. the X-axis, wrote "the size of the agglomerates and should write only “size of the
agglomerates”. The error has been corrected in all figures.
Round 3
Reviewer 2 Report
The authors have addressed all the points that were found to improve the manuscript, and I recommend that it be accepted for publication.